# Improving Thickness Uniformity of Mo/Si Multilayers on Curved Spherical Substrates by a Masking Technique

**Zhe Zhang** [1,2], **Runze Qi** [1,2], **Yiyun Yao** [1,2], **Yingna Shi** [1,2], **Wenbin Li** [1,2], **Qiushi Huang** [1,2], **Shengzhen Yi** [1,2], **Zhong Zhang** [1,2], **Zhanshan Wang** [1,2] and **Chun Xie** [3,*]

[1]   MOE Key Laboratory of Advanced Micro-Structured Materials, No.1239 Siping Road, Shanghai 200092, China; zzfight1226@foxmail.com (Z.Z.); qrz@tongji.edu.cn (R.Q.); yyy_1117@163.com (Y.Y.); shiyingna192514@126.com (Y.S.); wbli@tongji.edu.cn (W.L.); huangqs@tongji.edu.cn (Q.H.); 023123@tongji.edu.cn (S.Y.); zhangzhongcc@tongji.edu.cn (Z.Z.); wangzs@tongji.edu.cn (Z.W.)

[2]   Institute of Precision Optical Engineering, School of Physics Science and Engineering, Tongji University, Shanghai 200092, China

[3]   Sino-German College of Applied Sciences, Tongji University, Shanghai 200092, China

*   Correspondence: xc0522@tongji.edu.cn; Tel.: +86-21-6598-4734

**Abstract:** In this work, a masking technique was used to improve the thickness uniformity of a Mo/Si multilayer deposited on a curved spherical mirror by direct current (DC) magnetron sputtering with planetary rotation stages. The clear aperture of the mirror was 125 mm with a radius of curvature equal to 143.82 mm. Two different shadow masks were prepared; one was flat and the other was oblique. When using the flat mask, the non-uniformity considerably increased owing to the relatively large gap between the mask and substrate. The deviation between the designed and measured layer thickness and non-uniformity gradually reduced with a smaller gap. The second mask was designed with an oblique profile. Using the oblique mask, the deviation from multilayer thickness uniformity was substantially reduced to a magnitude below 0.8% on the curved spherical substrate over the clear aperture of 125 mm. Multilayers still achieved a smooth growth when deposited with obliquely incident particles. The facile masking technique proposed in this study can be used for depositing uniform coatings on curved spherical substrates with large numerical apertures for high-resolution microscopes, telescopes, and other related optical systems.

**Keywords:** layer uniformity; magnetron sputtering; shadow mask; multilayer; curved substrate

## 1. Introduction

Extreme ultraviolet (EUV) light sources, such as free electron lasers (FELs) [1,2], high harmonic generation devices [3], and plasma-based sources [4,5], produce ultra-short radiation with extremely high intensity. During practical utilization, the optical components and detectors of such sources are inevitably subjected to powerful radiation, which necessitates thorough investigation of their damage resistances. As the beamtime provided by FEL facilities is limited, special laboratory equipment is used to conduct damage tests on optical elements in the short-wavelength region. High-numerical-aperture (high-NA) optical systems are most suitable for performing such tests in a laboratory owing to the high fluence of EUV radiation [6,7]. Using the simplest high-NA optical system consisting of two mirrors [7], a tabletop EUV focusing optical system with an operational wavelength of 13.5 nm was developed at the Institute of Precision Optical Engineering (IPOE) for investigating the damage of optical elements caused by EUV radiation [8]. This optical system contains a modified Schwarzschild objective [5] with an NA of 0.44 and two mirrors with the same radius of curvature (RoC) equal to 143.82 mm. Because

of the high NA, one mirror (called a secondary mirror) is concave and has a clear aperture (CA) of 125 mm, while the other (primary) mirror is convex, and its CA is 56 mm. Uniform deposition of coatings on these mirrors is required to achieve constant optical properties over the optical element surface. If the degree of coating uniformity is not sufficiently high, the resulting non-uniform thickness profiles may seriously affect the efficiency of the entire system. Therefore, controlling the thickness uniformity of coatings is critical for proper focusing of tabletop EUV optical systems (especially for the curved substrates similar to the secondary mirror).

Typically, direct current (DC) magnetron sputtering systems with planetary rotation stages (PRS) [9] produce uniform coatings on flat substrates, but they cannot achieve a high degree of thickness uniformity in coatings deposited on curved substrates. This problem is solved by changing the speed profile of substrate motion [10–12], which is equivalent to controlling the deposition rate of sputtered materials at different positions along the radial axis [10–17]. Yu et al. utilized this technique to reduce the thickness deviation of the Mo/Si multilayer deposited onto a curved mirror with a CA of 184 mm and RoC of 338.66 mm through DC magnetron sputtering with PRS below 0.09% [12]. Further, the speed profile of substrate motion is always controlled during the deposition of optical components for EUV lithography (EUVL) [10–13]; however, the utilized deposition systems must possess extremely high mechanical accuracy, which is very difficult to achieve under laboratory conditions. To mitigate this issue, shadow masks are typically used for coating uniformity optimization via the selective blocking of sputtered materials [18–29]. Nevertheless, there are very few existing studies on the application of shadow masks to material deposition on the surfaces of curved substrates (such as secondary mirrors). Because the majority of the shadow masks used in DC magnetron sputtering are flat and mounted parallel to the sputtering source, their utilization may increase the gap between the mask and a curved substrate, thus causing a serious shadow effect. Therefore, a novel masking technique must be considered for depositing uniform coatings on curved substrates through DC magnetron sputtering with PRS.

In this study, a simple and facile method for designing shadow masks was developed at the IPOE to deposit uniform multilayers on curved spherical substrates through DC magnetron sputtering with PRS, which did not require consideration of the ejection characteristics of different sputtering materials [30,31]. The multilayer structure, utilized DC magnetron sputtering system, thickness measurement method, and mask design are described in detail in the subsequent sections. Using this technique, uniform multilayers were obtained on the surfaces of secondary mirrors.

## 2. Experiments

### 2.1. Multilayer Design

The modified Schwarzschild objective consists of on-axis spherical primary and secondary mirrors with the same RoC of 143.8 mm that provides a clear annular aperture (see Figure 1). The centers of these mirrors contain holes for light rays with diameters of 13 and 35 mm, respectively. As the working wavelength of the modified Schwarzschild objective is 13.5 nm, molybdenum/silicon (Mo/Si) multilayers must be deposited on the mirrors. According to Bragg's law with refractive correction, the range of incidence angles on the mirror surface is a critical parameter for multilayer design. Compared to the classical Schwarzschild objective [32,33], the modified objective exhibits smaller variations in incident angles, especially for the secondary mirror. In this case, geometrical ray tracing provides ranges of the incidence angles for different rays, which are equal to $3.31° < \theta < 11.08°$ and $1.18° < \theta < 3.84°$ for the primary and secondary mirrors, respectively.

As indicated by these ranges, the deposited Mo/Si multilayers should possess relatively high reflectivity with a periodic thickness based on Bragg's law with refractive correction. Figure 2 presents the theoretical reflectivities of the periodic Mo/Si multilayers optimized at a working wavelength of 13.5 nm. The d-spacing values of the multilayers deposited on the primary and secondary mirrors are equal to 7.03 and 6.91 nm, respectively. The thickness ratio of Si was set to 0.6 ($d_{si}/d$), and the interface

width was set to 0.3 nm. Moreover, the reflectivity bandwidth of the Mo/Si multilayer is relatively narrow; therefore, to achieve bandwidth matching with the primary mirror and obtain a high energy density at the focal plane, the deviation of the coating thickness uniformity over the clear aperture of the secondary mirror could not exceed 1%. In addition, the total number of bilayers was set at 30, which was also used for bandwidth matching between the two mirrors.

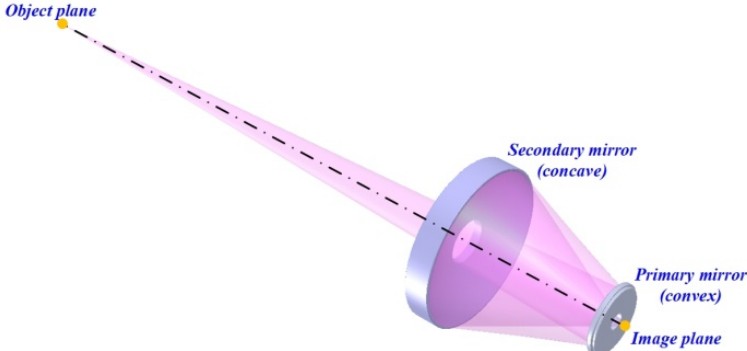

**Figure 1.** Schematic of the modified Schwarzschild objective.

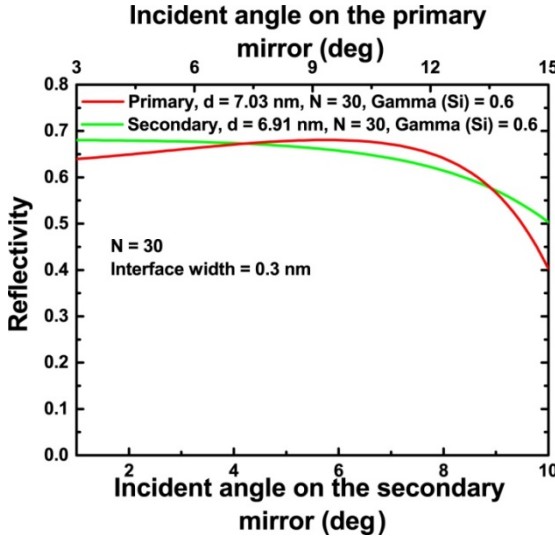

**Figure 2.** Reflectivities of the Mo/Si multilayers designed for the mirrors of the modified Schwarzschild objective.

## 2.2. Direct Current (DC) Magnetron Sputtering System

Multilayers were deposited through magnetron sputtering inside a high-vacuum chamber with a diameter of 1.20 m and height of 0.65 m. It consisted of four planar DC magnetron sources pointing upward and a sample holder for substrate mounting pointing downward. The target had a width of 128 mm and length of 381 mm. The sample holder was attached to a rotatable swivel plate. In the rotational mode, the sample swept past the target via orbital revolution. Successive layers were deposited above each target at a distance of 11.5 cm, as shown in Figure 3. Unlike controlling the layer thickness by varying the resting time over the respective target, the method for controlling the layer thickness in this study depended on the speed of sample movement over the target. The substrate rotated around the sample holder's axis of symmetry at a speed of 100 rpm to achieve good radial uniformity. In all experiments, high-purity argon (99.999%) was used as the working gas, the background pressure was $2.0 \times 10^{-4}$ Pa, and the argon gas pressure during deposition was 0.133 Pa.

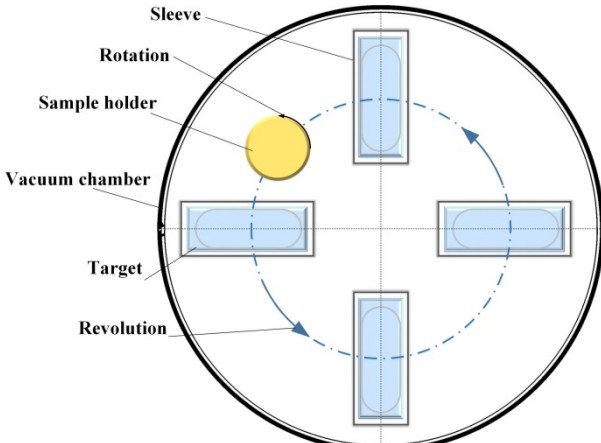

**Figure 3.** Schematic of the direct current (DC) magnetron sputtering system used in this study (top view).

As shown in Figure 2, multilayers can exhibit high reflectivity with a periodic thickness; however, for the deposition system used in our laboratory, the coating uniformity varied significantly because of the distance changed between the targets and mirrors along the radial axis, which ranged between 0 and 2.22 mm for the primary mirror and between 0 and 13.22 mm for the secondary mirror. According to the results of previous experiments, reaching a high degree of multilayer uniformity is much harder for the secondary mirror than for the primary mirror under the given conditions. Because the distance change between the target and primary mirror is small, the latter can be considered a flat substrate with relatively uniform multilayers. Therefore, the present study focused on controlling the coating uniformity on the secondary mirror. This task cannot be performed by controlling the speed profile during substrate motion because such a curved concave mirror undergoes a drastic speed change upon entering the sputtering area; moreover, its acceleration must be as small as possible, which is beyond the capability of our deposition system. Therefore, using a shadow mask is suitable for improving the thickness uniformity of the coatings.

*2.3. Mirror Substitute for Thickness Measurements*

The mask used in our experiments was designed using the following steps. First, the coating thickness was measured at several points. Second, the coating thickness distribution was obtained from the measurement data. Finally, several masks were prepared based on the fitting results. Because the coating thickness could not be directly measured on the secondary mirror, a mirror substitute was prepared for estimating its value at selected points on the coating surface under rotating conditions.

To reproduce the real curvature of the substrate as accurately as possible, eight points between the center of the secondary mirror and its boundary were selected, and their tangent line was used to create a cross-sectional line, as shown in Figure 4a. Here, the upper image shows the cross-section of the secondary mirror, and the blue spots represent the measurement points. Actually, the eight selected points are in one direction along the radial axis. To provide space for mounting silicon pieces on the mirror substitute, four points (dashed circles) are set symmetrically along the same direction, as shown in Figure 4a. The lower image depicts the cross-section of the mirror substitute; the red line denotes the silicon piece, and the angles correspond to the angles between the tangent lines of the selected points and the horizontal line. Based on this cross-sectional line, a surface capable of accommodating eight silicon (100) pieces with sizes of 20 mm × 10 mm × 0.5 mm was designed. By measuring the coating thickness of the silicon pieces, it was possible to determine its distribution on the substrate (a photograph of the mirror substitute and silicon pieces is shown in Figure 4b).

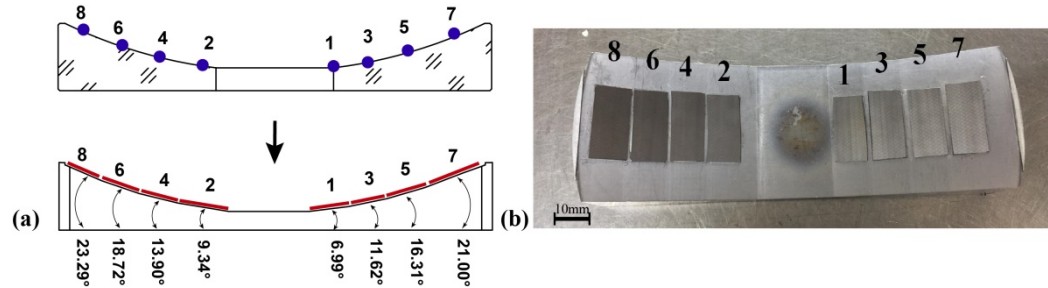

**Figure 4.** (**a**) Schematic illustration of the mirror substitute design (upper image is related to the secondary mirror and the lower one to the mirror substitute); (**b**) mirror substitute containing eight silicon pieces.

The fabricated Mo/Si multilayers were characterized via grazing incidence X-ray reflectometry (GIXR) using Cu-K$\alpha$ radiation ($\lambda$ = 0.154 nm) and a laboratory diffractometer (D1 system, Bede Inc.). Silicon piece 1 depicted in Figure 4b is located very close to the hole of the secondary mirror and exhibits some signs of edge damage caused during fabrication and substrate polishing. Therefore, this piece was not used for thickness measurements. The remaining silicon pieces were examined at their central parts along the short edges.

### 2.4. Shadow Mask Design

In this study, a shadow mask was fixed onto the substrate holder to improve the coating thickness uniformity. To achieve this goal, the substrate had to possess revolution symmetry and the shadow mask had to be as close to the substrate as possible. During sputtering, the particles emitted from the deposition target flew through the fixed mask. As shown in Figure 5, at a given radius $r$, the mask covers the angular range $[-\theta(r), \theta(r)]$, and the total layer thickness decreases with substrate rotation. The mask profile at this radius can be described by the following equation:

$$x = r\cos(\theta(r)), \quad y = r\sin(\theta(r)) \tag{1}$$

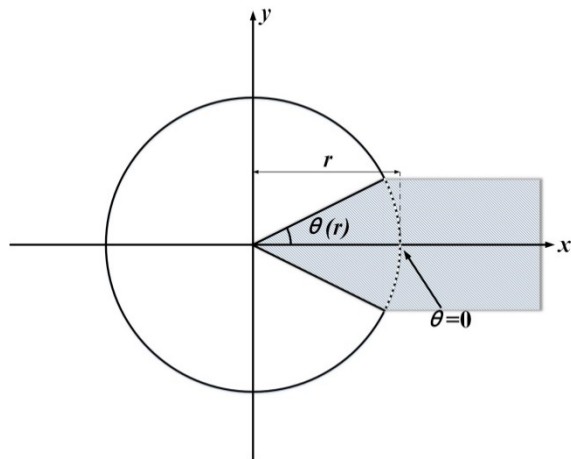

**Figure 5.** At radius $r$, the mask (grey shaded area) blocks particles flying in the angular range $[-\theta(r), \theta(r)]$.

Therefore, the most important step in designing the mask was to determine the function $\theta(r)$ via the following equation:

$$\theta(r) = \frac{180°}{M} \times \left[\left(1 - \frac{d_{idea}}{d_{start}(r)}\right)\right] \tag{2}$$

where $d_{idea}$ is the desired coating thickness, and $d_{start(r)}$ is the coating thickness at radius $r$ on the unmasked rotary substrate. $M$ is the number of mask segments (there are eight segments in a mask), which is inversely proportional to the segment width.

In this work, the distribution of the coating thickness was obtained as follows. First, the multilayer thickness on the mirror substitute was determined. Second, it was assumed that the obtained data represented the coating thicknesses of a projection plane measured at some points, as indicated by the double arrows between the projection plane and cross-section of the secondary mirror in Figure 6. Hence, the thickness distribution across this plane could be obtained by fitting. Finally, it was possible to determine the profile of the mask parallel to the projection plane using Equation (2). By utilizing this method, two different shadow masks were designed. As shown in Figure 6, mask #1 is prepared based on the thickness distribution obtained for the plane parallel to the sputtering source surface, and mask #2 is designed based on the thickness distribution obtained for the plane passing at an angle of 16.38° with respect to the sputtering source surface. The coating thickness distribution was determined by repeating the deposition procedure several times for better accuracy. In addition, the value of $M$ in our design was eight.

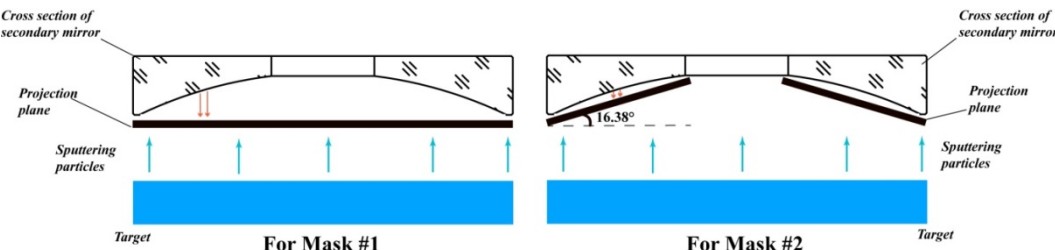

**Figure 6.** Schematics of the coating thickness distributions on different projection planes obtained for masks #1 and #2.

## 3. Results and Discussion

The multilayer thickness distribution on a rotating substrate was determined by measuring the multilayer thicknesses on the silicon pieces depicted in Figure 4b. Here, the position of the silicon piece whose thickness was the highest was taken as the reference point. The normalized thickness for each point was obtained by dividing the thickness at that point by the thickness at the reference point. The normalized thicknesses measured via GIXR along the radial axis are presented in Figure 7. Based on the repeatability of deposition and measurement, the error bar shown in Figure 7 is ±0.2%. Then, by projecting the normalized coating thickness to different planes used for different mask designs, the different thickness distributions will be obtained. Next, the obtained coating thickness distribution was used to calculate the profile of the mask using MATLAB software [34].

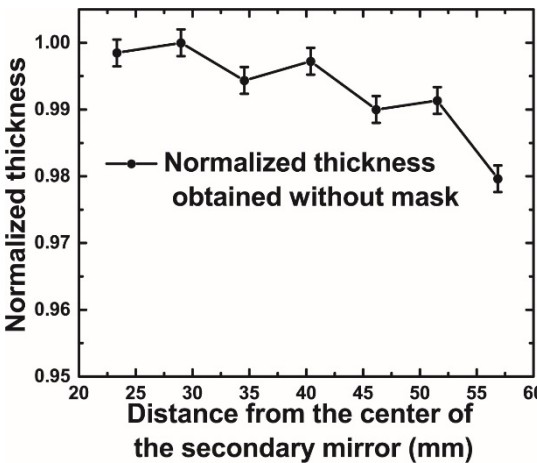

**Figure 7.** Normalized thickness versus the distance from the center of the secondary mirror obtained without mask.

The profile of each mask segment calculated from the thickness distribution is shown in Figure 8 (note that the segments of mask #2 are 0.6 mm narrower and approximately 2 mm longer than those of mask #1). The fabricated masks are shown in Figure 9a; the holes near the mask boundary were designed to fix the mask on the sample holder. To correctly align the substrate and mask, they were mounted on the same axis. During deposition, the substrate was rotated, whereas the mask was fixed on the sample holder. The assembly containing the secondary mirror and mask #2 is displayed in Figure 9b. Apart from the shape of the segments, another difference between the two masks is the distance between the mask and substrate because of the different profiles of the two masks. The distance between mask #1 and the substrate ranged from 12.38 mm to 2.57 mm from the center to the edge of the substrate. For mask #2, the distance between the mask and substrate was lower than 2.50 mm on the whole area.

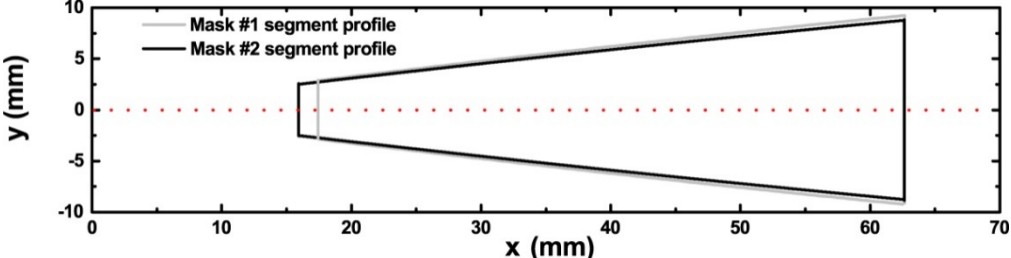

**Figure 8.** Segment shapes obtained for masks #1 and #2.

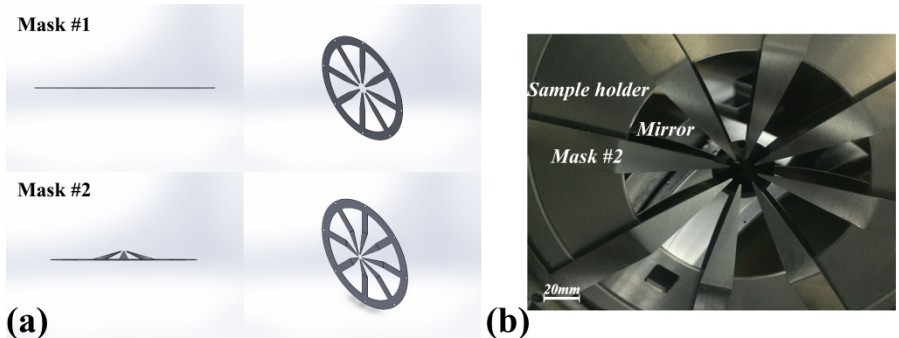

**Figure 9.** (**a**) Images of the fabricated masks (left images correspond to side views, and right images correspond to isometric views); (**b**) mounting on the sample holder with the secondary mirror and mask #2.

Figure 10a shows the normalized multilayer thicknesses on the secondary mirror plotted as functions of the distance from the mirror center to the edge, obtained without the mask, with mask #1 and mask #2. The nonuniformity of the coating thickness was reduced to below 0.8% when mask #2 was used during deposition. Without any mask and with mask #1, the non-uniformities of the coating thickness are much higher, at 2% and 17%, respectively. A more detailed comparison is presented in Figure 11, showing the GIXR results of the multilayers fabricated using both masks at the same positions (silicon pieces 2, 5, and 8). At position 2, the d-spacing of the multilayer fabricated by mask #1 is only 5.41 nm, which is much smaller than 6.93 nm (d-spacing of multilayer fabricated using mask #2). Moreover, the Bragg peaks obtained with mask #1 are much wider than those obtained with mask #2. Because the characterization conditions were the same for all these samples, the most likely reason for wider Bragg peaks is the higher lateral gradient of the d-spacing. At position 2, the distance between mask #1 and the substrate is 12.38 mm and this distance is 0.88 mm for mask 2. At position 5, the d-spacing of the multilayer deposited with mask #1 is 6.35 nm, which is higher than that at position 2. It is closer to the value obtained using mask #2 at the same position, compared to that at position 2. The Bragg peaks also become narrower than the one at position 2 but still wider than that for the multilayer deposited using mask #2. At this position, the distance between mask #1 and the substrate is 8.50 mm, and that between mask # 2 and the substrate is 1.52 mm. At position 8, the d-spacing of the multilayer obtained with mask #1 is 6.52 nm, which is very close to the value obtained with mask #2 compared to that at other positions. In addition, the widths of the Bragg peaks obtained using the two masks are also similar. At this position, the distances between masks and the substrate are close to each other (2.57 mm vs. 2.01 mm corresponding to mask #1 vs. mask #2). Based on this, the d-spacing values of all samples fabricated using mask #1 are presented in Figure 12. The plot shows the d-spacing of the multilayers as a function of the distance between the mask and substrate, and the magenta dashed line represents the designed d-spacing. Noting that the designed uniform d-spacing values were achieved using mask #2, as depicted in Figure 10a. The experimental results indicated that the d-spacing was closer to the designed value when the gap (the distance between the mask and substrate) became smaller. By contrast, the d-spacings deviated considerably from the designed values when the gap was large. Based on these results, it can be concluded that the reason for the d-spacings of multilayers deposited with mask #2 being close to the designed value is the small gap between mask #2 and the substrate on the whole area. According to Figure 12, the gap between the mask and substrate should be kept within 5 mm to obtain the designed d-spacing values.

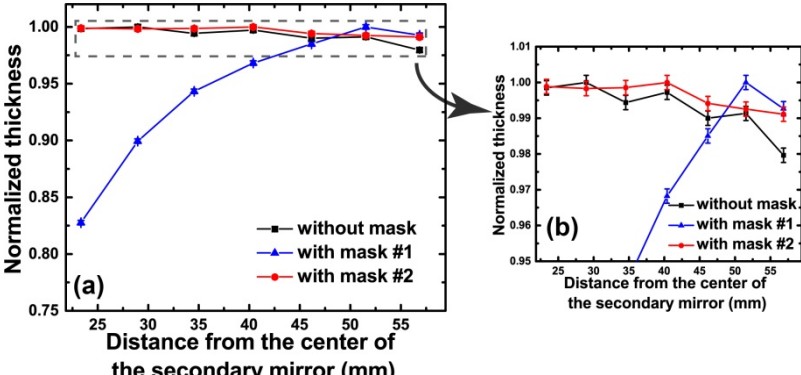

**Figure 10.** (**a**) Normalized thickness profiles obtained without mask, with mask #1, and with mask #2; (**b**) enlarged view of the region marked with grey dashed rectangle.

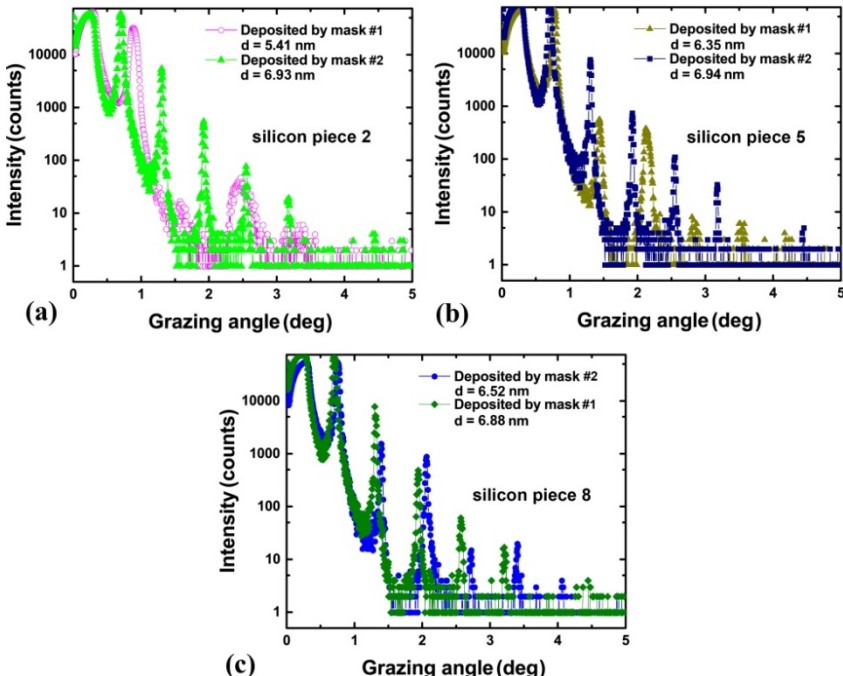

**Figure 11.** Grazing incidence X-ray reflectometry (GIXR) measurements of the multilayers deposited on different silicon pieces at the same positions using masks #1 and #2: (**a**) silicon piece 2; (**b**) silicon piece 5; (**c**) silicon piece 8.

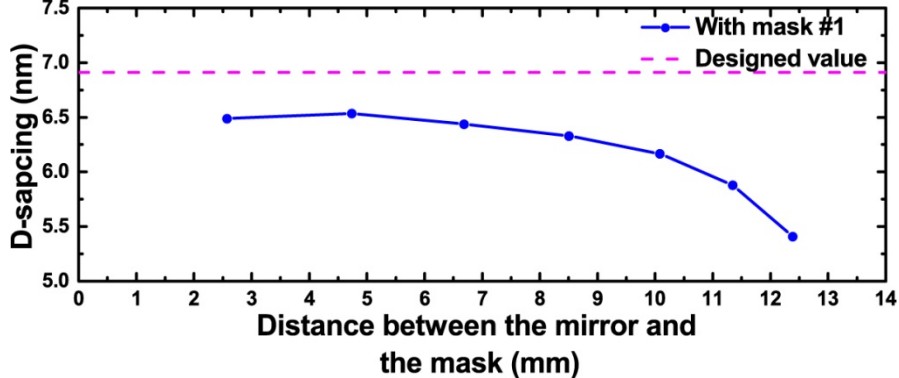

**Figure 12.** Multilayer d-spacing values as functions of the distance between the mirror and mask in case of mask #1. The magenta dashed line represents the designed value of d-spacing.

The reason for the deviation between the designed value and the deposited layer thicknesses using mask #1 can be further analyzed as follows. When designing the mask, it was assumed the mask needed to be mounted as close as possible to the substrate to block a portion of the sputtering particles. However, for mask #1, the gap between the mask and substrate at the center position of the substrate was considerably large and changed substantially from the center to the edge. In this case, the proportion of the sputtered particles blocked by mask #1 deviated from the designed model and this deviation significantly changed with the size of the gap. As shown in Figure 13, taking an extended sputtering source with a divergent sputtering area as an example, when a mask is inserted to block a portion of the sputtering particles, the size of the blocked area will change with changes in the distance between the sputtering source and substrate [35]. If there was a larger gap between the mask and substrate, the blocked area would become bigger, and the mask would block more particles resulting in a smaller thickness. As shown in Figure 12, the gap at position 2 is the largest, which makes the d-spacing of the deposited multilayers significantly smaller than the designed value. Moreover, the gap quickly reduced from position 2 to position 5, which brought a significant

change on the blocking effects and different deviations of the deposited d-spacing from the designed constant value; this resulted in a large lateral gradient on this area and caused broadening of the Bragg peaks. At positions 5 and 8, the gap became smaller, and the changes in the d-spacing also reduced; this resulted in the real deposition geometry becoming closer to the assumed model, and thus the d-spacing of the deposited multilayers became closer to the designed value. These findings are consistent with the GIXR measurements presented in Figure 11.

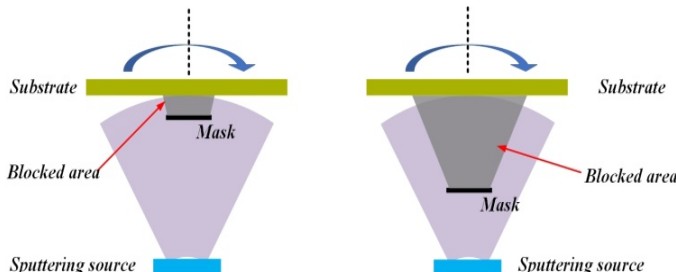

**Figure 13.** Schematic illustration showing changes in the blocked area with different gaps between mask and substrate.

The parameters of the deposited Mo/Si multilayers were fitted using the Bede REFS software package [36], and the fitting model was based on previous research [37]. Figure 14 shows the GIXR curves and fitted results of the Mo/Si multilayers (deposited with mask #2) at different positions (silicon pieces 2, 5, and 8). As shown in Figure 14, the fitted curves for the samples match very well with the GIXR measurement results. The fitted d-spacing values and angles between the sample surface and sputtering target surface (represented by $\alpha$) for each sample are also presented in the plot. Interface widths between 0.25 and 0.50 nm were observed, which agreed with previously reported results [38].

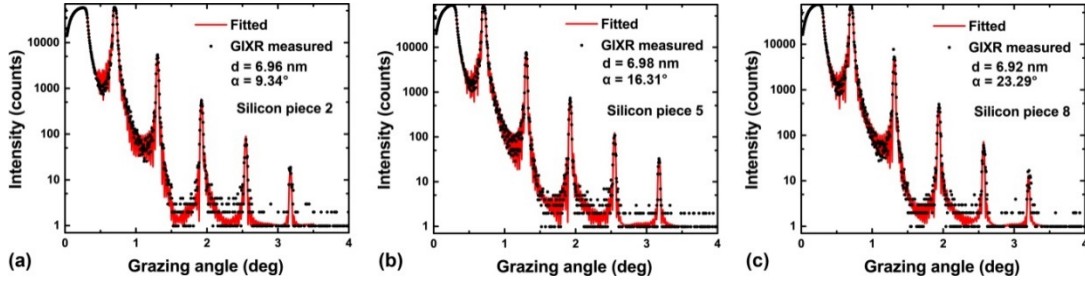

**Figure 14.** GIXR measurement results (black dots) and fitted (red line) GIXR curves of the multilayers deposited with mask #2 on different silicon pieces: (**a**) silicon piece 2; (**b**) silicon piece 5; (**c**) silicon piece 8. (Fitted d-spacings and angles between the sample surface and sputtering target surface are shown in the plots.).

Another important fact is that these multilayers were deposited on a surface that was not parallel to the sputtering target surface. From a theoretical viewpoint, deposition using different $\alpha$ values will influence the formation of the multilayers [39]. Voronov et al. investigated how different $\alpha$ values influenced the interface of Mo/Si multilayers formed using ion beam sputtering (IBS). The results of their work indicated that the interface changed considerably when $\alpha$ was larger than 45° [40]. In this work, the divergence angle of the sputtering particles was higher than that in IBS, which may cause some difference on this issue. According to the fitted results, the multilayers deposited with $\alpha$ values of up to 20° could have similar interface structures compared to the multilayers deposited with $\alpha = 0°$, i.e., when the substrate is parallel to the sputtering target surface. Therefore, from 0° to 20°, changes in $\alpha$ will not significantly influence the formation of high-quality multilayers with smooth interfaces under magnetron sputtering. As all the optical mirrors in high-NA systems are used for performing

depositions with different $\alpha$ values, this result can provide useful guidance for effective multilayer deposition on such curved optical surfaces.

Using the secondary mirror deposited with mask #2, the modified Schwarzschild showed good performance. The highest EUV energy density acquired by this optical system was $2.27 \pm 0.27$ J/cm$^2$. Such a high EUV energy density results from the satisfactory performance of such mirrors, and this value is approximately two times higher than previously reported results [5]. The objective with a mechanical mount and the damage profile of a gold mirror are shown in Figure 15. In summary, this masking technique is a promising method to deposit uniform multilayers on similar mirrors used in other high-NA optical systems like microscopes, lithography systems, and telescopes.

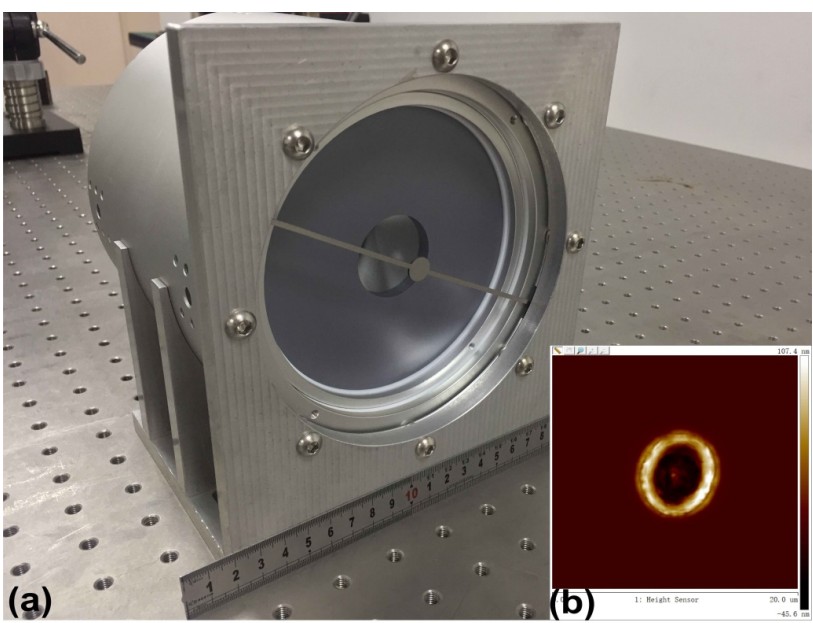

**Figure 15.** (**a**) Modified Schwarzschild objective with mechanical mount after completing the alignment; (**b**) atomic force microscopy (AFM) image of a gold mirror irradiated with a single pulse.

## 4. Conclusions

A new masking technique was developed for improving the thickness uniformity of deposited Mo/Si multilayers. In particular, two different masks were prepared to enhance the coating thickness distribution on a spherical concave substrate with CA/RoC = −0.87. The multilayer thickness on the mirror substrate was measured several times to ensure good reproducibility of the results obtained; these results were subsequently used to determine the multilayer thickness distribution on a real mirror. The mask shape was designed using MATLAB software. By comparing the results obtained from depositions using these two different masks, it was found that the mask mounted as close as possible (a distance of approximately 5 mm between the substrate and mask) to the substrate could efficiently help to control the thickness uniformity of the multilayers. The deviation of the coating thickness decreased to below 0.8% of the designed value using the developed mask combined with the PRS of the DC magnetron sputtering system. The multilayers showed smooth growth when deposited at different angles ($0° < \alpha < 20°$). Thus, in this study, we developed a simple, fast, and efficient masking technique that is easy to implement at low manufacturing costs. Using this technique, high uniformity of multilayers can be achieved on a substrate with any rotational symmetry. In addition, the results of multilayer deposition on a surface not parallel to the sputtering target surface can provide guidance for fabricating other curved optics using similar methods.

**Author Contributions:** Conceptualization, Z.Z. (Zhe Zhang), Z.W. and C.X.; methodology, Z.Z. (Zhe Zhang), R.Q.; software, Y.Y., Y.S.; validation, R.Q., Z.Z. (Zhong Zhang), and Q.H.; formal analysis, Z.Z. (Zhe Zhang), R.Q., S.Y. and Q.H.; investigation, Z.Z. (Zhe Zhang), R.Q., and Y.S.; resources, Q.H., Z.Z. (Zhong Zhang); data curation,

Z.Z. (Zhe Zhang), Y.S. and R.Q.; writing—original draft preparation, Z.Z. (Zhe Zhang), R.Q., Q.H.; writing—review and editing, R.Q., Q.H., Z.W. and C.X.; visualization, Z.Z. (Zhe Zhang); supervision, Q.H., Z.W., and Z.Z. (Zhong Zhang); project administration, C.X.; funding acquisition, Z.W., W.L.

**Funding:** This work is supported by the National Key R&D Program of China (grant No. 2016YFA0401304) and National Natural Science Foundation of China (NSFC; grant Nos. 11875203 and 61621001).

**Conflicts of Interest:** The authors declare no conflict of interest. The funders had no role in the design of the study; in the collection, analyses, or interpretation of data; in the writing of the manuscript, or in the decision to publish the results.

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
