# Peer review of "Improving Thickness Uniformity of Mo/Si Multilayers on Curved Spherical Substrates by a Masking Technique"

_coatings, doi:10.3390/coatings9120851_

Round 1

Reviewer 1 Report

The article presents the study of a masking technique that can be used for depositing uniform coatings on curved spherical substrates with large numerical apertures for high-resolution optical systems. As a proof of concept, the Mo/Si multilayer was analysed. The manuscript is well written, experimental part contains all necessary details to address the analysis, the experiments are properly planned and the conclusions are well supported by the obtained data. Therefore, I recommend to accept this manuscript for publication in Coatings as submitted.

One remark for Authors:

- The first paragraph in the Introduction: “The introduction should briefly place the study in a broad context and highlight why it is 31 important. It should define the purpose of the work and its significance. The current state of the 32 research field should be reviewed carefully and key publications cited. Please highlight controversial 33 and diverging hypotheses when necessary. Finally, briefly mention the main aim of the work and 34 highlight the principal conclusions. As far as possible, please keep the introduction comprehensible 35 to scientists outside your particular field of research. References should be numbered in order of 36 appearance and indicated by a numeral or numerals in square brackets, e.g., [1] or [2,3], or [4–6]. See 37 the end of the document for further details on references.” should be deleted.

Author Response

Thank you for your careful review. We have deleted this mistake in revised manuscript.

Reviewer 2 Report

The article is focused on the on the use of shadow mask in place of speed control in order to obtain homogeneous coatings on curved substrates. Speed control in Mo/Si has already confirmed to allow thickness deviation below 0.09%, however it requires high mechanical accuracy of the system. The mask is easier to implement and the authors achieved thickness deviation below of 0.8%. The result is an order of magnitude higher than in speed control but it can be enough for some optics.

The masking method and the model elaborated is simple, more than a scientific advance is more a technical report. Nevertheless can be of interest for the community. In order to publish I suggest some major revisions:

1. delete from line 31-28

2. when there are photos in pictures as in Fig.4 add a scale bar.

3. Fig. 7 the data need an error, please provide it. It seems that all the even points are high than odd ones, suggesting a systematic error in the orientation or in the high in the dummy mirror . Explain and write how the whole (systematic and random) error have been calculated. How many samples were produced to have the profile? and how many dummy mirrors? What is the accuracy in the positioning? The fluctuation between to following point suggests that this error is not negligible.

4. Fig. 10 add the error bars. It cannot be understood if the differences are significative or not. It is not only the error in the XRR, but also the one of positioning horizontally and vertically. Explain and write how the whole (systematic and random) error have been calculated.

4. line 279-285 the discussion is quite confused and in particular those lines seem a repetition. Please reorder the full paragraph on the discussion of the thickness profiles.

5. line 319-320 please expand explaining better setup and meaning.

Round 2

Reviewer 2 Report

The authors have considered all the suggestions. In particular with error bars it appears clear that the improvement between with and without mask is only for for radius larger than 50 mm.

The masking method and the model elaborated is simple, more than a scientific advance is more a technical report. Nevertheless I suggest to publish it in the present form.